# Accelerating the decentralized federated learning via manipulating edges in complex graph

## ABSTRACT

Federated learning enables collaborative AI training across organizations without compromising data privacy. Decentralized federated learning (DFL) improves this by offering enhanced reliability and security through peer-to-peer (P2P) model sharing. However, DFL faces challenges in terms of slow convergence rate due to the complex P2P graphs. To address this issue, we propose an efficient algorithm to accelerate DFL by introducing a limited number $k$ of edges into the P2P graphs. Specifically, we establish a connection between the convergence rate and the second smallest eigenvalue of the laplacian matrix of the P2P graph. We prove that finding the optimal set of edges to maximize this eigenvalue is an NP-complete problem. Our quantitative analysis shows the positive effect of strategic edge additions on improving this eigenvalue. Based on the analysis, we then propose an efficient algorithm to compute the best set of candidate edges to maximize the second smallest eigenvalue, and consequently the convergence rate is maximized. Our algorithm has low time complexity of $O(krn^2)$. Experimental results on diverse datasets validate the effectiveness of our proposed algorithms in accelerating DFL convergence.

## CCS CONCEPTS

• **Computing methodologies → Distributed computing methodologies**.

## KEYWORDS

Decentralized federated learning, convergence rate, influential edge

**ACM Reference Format:**
Anonymous Author(s). 2018. Accelerating the decentralized federated learning via manipulating edges in complex graph. In *Proceedings of Make sure to enter the correct conference title from your rights confirmation emai (Conference acronym 'XX).* ACM, New York, NY, USA, 10 pages. https://doi.org/XXXXXXX.XXXXXXX

## 1 INTRODUCTION

Artificial intelligence (AI) has found applications in diverse domains, such as natural language processing, face recognition, recommender systems, and smart transportation. However, due to privacy concerns and commercial competition, certain data cannot be shared or moved among different organizations or agents [1]. To address this challenge, Federated Learning (FL) has emerged as a promising approach that enables multiple agents to collaboratively train AI models while preserving the privacy of local data [2]. In classical federated learning, each agent trains a local model and shares its model parameters (or parameter gradients) with a central parameter server. The server then updates the model and sends it back to the agents. However, this approach is hindered by communication and computational bottlenecks of the central server with the number increase of agents [1]. Additionally, in situations such as wireless sensor networks, some agents may not have direct access to the central parameter server. As a result, Decentralized Federated Learning (DFL) has garnered significant attention in recent years [3, 4]. This approach eliminates the need for a centralized parameter server and allows agents to exchange model parameters solely with their neighboring agents. By doing so, the decentralized framework mitigates the bandwidth requirements for the server and presents a more scalable and efficient solution for systems with millions(or more) of agents [5].

In DFL, ensuring model consensus among different agents is a prerequisite. The consensus criteria in the fields of decentralized learning and complex networked systems is commonly characterized by the ratio of the second smallest eigenvalue to the largest eigenvalue of the laplacian matrix of a graph [4, 6, 7]. An interesting and more general problem emerges: how to enhance the convergence rate of DFL. This problem is closely related to various factors such as the computation speed of agents, the bandwidth between agents, the communication protocol, and the graph topology [4, 5]. Over past decades, the sizes of deep learning models have been rapidly growing, reaching up to more than 100MB. Frequently transferring such large model parameters, even with model compression, imposes a significant burden on the bandwidth [5]. To address this issue, designing an appropriate graph topology to expedite convergence and reduce the frequency of model exchange between agents becomes a plausible solution [4, 8]. On one hand, previous research has shown that many real-world large graphs, including social/mobilephone networks with over 1 million users, exhibit remarkably slow convergence rates [9], implying that we should add more edges to accelerate the convergence rate. On the other hand, certain edges in the graph may contribute minimally, or not

at all, to the model consensus, implying that we should remove these edges to avoid insignificant communication. To the best of our knowledge, there is a scarcity of literature that quantitatively analyzes the optimization of graph topology associated with convergence rate in DFL.

In this paper, we propose a novel approach, namely Second Smallest Eigenvalue Optimization (SSEO) and its variant SSEO+, to enhance the convergence rate of DFL by incorporating a budget number $k$ of additional edges. We first show that accelerating DFL could be achieved by increasing the second smallest eigenvalue of the Laplacian matrix associated with a graph, and how determining the best set of edges is an NP-complete problem. Subsequently, we employ the eigenvalue perturbation technique to analyze the impact of edge addition on the second smallest eigenvalue. Finally, we devise an efficient algorithm to select the optimal set of edges to accelerate DFL. The primary contributions of this study can be summarized as follows:

- *Understanding Convergence Through Graph Metrics*: Our research explores the intricate relationship between model training convergence rate and the second smallest eigenvalue of the associated graph's Laplacian matrix. We extend this by investigating how edge manipulation–specifically, the addition or removal of edges–affects this critical eigenvalue.
- *Design new Topology Optimization Algorithms*: We introduce two novel algorithms, SSEO and its enhanced version SSEO+. These algorithms are designed to select the most beneficial set of edges that maximize the second smallest eigenvalue, serving as a pivotal factor for faster convergence. Notably, SSEO operates with a time complexity of $O(kn^2)$ while SSEO+ features a time complexity of $O(krn^2)$.
- *Conduct extensive empirical validation experiments*: We conduct extensive experiments using various datasets. The results unequivocally demonstrate that SSEO and SSEO+ outperform state-of-the-art baselines.

## 2 RELATED WORK

**Federated learning**: FL was initially proposed by researchers at Google in 2016 [3]. FL operates by sharing model parameters rather than user data, thus ensuring privacy preservation and overcoming geographical limitations, enabling efficient collaboration on a global scale [2, 10, 11]. However, traditional central server-based FL faces challenges of communication and computational bottlenecks [2, 5]. As a result, DFL has garnered significant attention in recent years. Noteworthy studies in this area encompass peer-to-peer FL [12], server-free FL [8], serverless FL [13], device-to-device FL [14], swarm learning [15], among others [2]. In DFL, each agent interacts with its neighboring agents, leading to the formation of diverse structures, including line, ring, fully-connected, and complex graph structures [4]. Aysal et al. [16, 17] investigated the communication protocols. Additionally, Nedić et al.

[4] examine the influence of network topology on the problem. Furthermore, Yuan et al. [2] provided a comprehensive summary of recent works on DFL.

**Distributed Optimization**: The distributed optimization models aims to design enhance the performance of learning models in distributed environment. Tsitsiklis et al. [18–21] demonstrated that the decentralized gradient descent strategy achieves average consensus and presented convergence rate criteria for both static and time-varying graphs. Furthermore, Nedić et al. [22–24] investigated decentralized optimization of nondifferentiable (but convex) functions. Kempe et al. [25–27] extended decentralized optimization from directed graphs to directed graphs. Ram et al. [28–30] examined the influence of noise on the DFL. Additionally, He et al. [8] introduced DFL in single-sided trust social networks. It was shown by Nedić et al. [4] that a dense topology leads to a faster convergence rate, implying less training epochs, but it may result in network congestion and long time delays. To address this, Wang et al. [1, 31] proposed MATCHA, which divides the original topology into disjoint communication subgraphs, reducing the frequency of model transmission. Zhou et al. [32] accelerate decentralized training by assigning a high probability to high-speed links for peer communication.

**Edge manipulation**: Manipulating edges in the graph is a plausible solution to improve the convergence rate of DFL. This involves adding new edges to the graphs and removing redundant ones. The problem has been extensively studied in the context of synchronization (consensus) in complex networks [33, 34]. Various researchers have made significant contributions in this area. For instance, Pecora et al. [35, 36] explored synchronization criteria based on the eigenvalues of the graph's Laplacian matrix. They investigated how the addition, removal, and relocation of individual edges influenced the network's ability to synchronize. Similarly, Tong et al. [9, 37, 38] identified and removed the influential nodes to manipulate the eigenvalues of the adjacency matrix and Laplacian matrix respectively. Additionally, Hagberg et al. [6, 39] proposed a method to improve synchronization by rewiring the edges in the graph. The studied graphs encompass variations such as undirected/directed, unweighted/weighted, and time-varying properties [40–42].

Our study draws parallels with the edge rewiring optimization and influential edge detection in refs. [9, 39]. In these studies, the authors focused on rewiring edges to enhance synchronization. In contrast, our objective is to manipulate a budget number of influential edges to augment the convergence rate of DFL. Our work differs from previous studies in three key aspects: (a) We employ a distinct criterion to quantify the impact of edge addition on the second smallest eigenvalue. (b) We propose a rapid algorithm that guarantees the performance error, whereas previous methods were predominantly heuristic. (c) While prior works rewired edges to enhance synchronization, our algorithm concurrently adds new edges and removes redundant existing ones. To the best of our knowledge, research on edge manipulation specifically targeting DFL remains limited. Hence, our paper offers a viable way to increase the convergence rate of DFL.

# 3 NOTATIONS AND PROBLEM DEFINITIONS

In this paper, we focus on the Decentralized Federated Learning (DFL) involving a set of $n$ clients, where each client represents an edge server or another type of computing device (e.g., a mobile phone). Each client possesses local private data $\mathcal{D}_i$ and a local model $\mathbf{x}_i$, $i = 1, 2, ..., n$. The communication structure among these agents can be represented by an undirected graph $\mathbb{G} = (\mathbb{V}, \mathbb{E})$, which is conveniently represented by the adjacency matrix $A = (a_{ij})_{n \times n}$, where $a_{ij} = 1$ if node $i$ is connected to node $j$, and $a_{ij} = 0$ otherwise. Notably, each node can only transmit its model $\mathbf{x}_i$ to its neighbors. The objective of DFL is to leverage this decentralized nodes to collectively train a shared model $\mathbf{x}$ based on the joint dataset. Specifically, we typically optimize the objective function $F(\mathbf{x})$ to compute $\mathbf{x}$:

$$F(\mathbf{x}) = \frac{1}{n} \sum_{i=1}^{n} F_i(\mathbf{x}) = \frac{1}{n} \sum_{i=1}^{n} \frac{1}{|\mathcal{D}_i|} \sum_{s \in \mathcal{D}_i} l(\mathbf{x}; s), \qquad (1)$$

where $F_i(\mathbf{x})$ is the local objective function associated with node $i$ and $l(\mathbf{x}; s)$ is the loss function for a data sample $s$.

The most common algorithm to minimize $F(\mathbf{x})$ is the stochastic gradient descent (SGD). In DFL, we usually use vanilla decentralized SGD (DecenSGD) to optimize $F(\mathbf{x})$. Let $\mathbf{x}_i^{(t)}$ denote the machine learning model on node $i$ at time $t$. DecenSGD updates $\mathbf{x}_i^{(t)}$ as follows:

(1) Initially, all nodes have identical model parameters $\mathbf{x}_i^{(0)} = \mathbf{x}^{(0)}, i = 1, 2, ..n$.

(2) *Parallel local SGD*: Node $i$ computes the stochastic gradient with respect to its local model parameters $\mathbf{x}_i^{(t)}$: $g_i(\mathbf{x}_i^{(t)}) = \frac{1}{|\mathcal{B}|} \sum_{s \in \mathcal{B}} \nabla l(\mathbf{x}_i^{(t)}; s)$, where $\mathcal{B}$ is a min-batch of randomly samples from the local dataset $\mathcal{D}_i$. Node $i$ updates the local model as: $\mathbf{x}_i^{(t + \frac{1}{2})} = \mathbf{x}_i^{(t)} - \eta g_i(\mathbf{x}_i^{(t)})$, where $\eta$ is the learning rate.

(3) *Communication with neighbors*: Node $i$ sends its local parameter $\mathbf{x}_i^{(t + \frac{1}{2})}$ to its neighbors, denoted as $N_i$, and receives parameters from its neighbors $\{\mathbf{x}_j^{(t + \frac{1}{2})}\}_{j \in N_i}$.

(4) *Model fusion*: Node $i$ mixes the local model parameter with these of its neighbors using a weighted average scheme: $\mathbf{x}_i^{(t+1)} = \sum_{j \in N_i \bigcup \{i\}} w_{ji} \mathbf{x}_j^{(t + \frac{1}{2})}$, where $w_{ji}$ is the $(j, i)-th$ entry of the mixing matrix $W = (w_{ji})_{n \times n}$.

(5) *Iteration*: Iteratively run the steps 2–4 until the $\mathbf{x}_i^{(t)}$ converges to a predefined error.

The overall update rule of DecenSGD follows as:

$$\mathbf{x}_i^{(t+1)} = \sum_{j=1}^{n} w_{ji} [\mathbf{x}_j^{(t)} - \eta g_j(\mathbf{x}_j^{(t)})]. \qquad (2)$$

The convergence of DecenSGD has been extensively explored in the field of decentralized learning. When analyzing the convergence property, we usually assume that the objective function is convex and the gradient norm is bounded (although specific assumptions may differ across studies). Based on these common assumptions, we ususally have the following convergence lemma:

**Lemma 1** (**Convergence property of DecenSGD** [43, 44]). Let $\bar{\mathbf{x}}$ denote the averaged model of all $\mathbf{x}_i$. After $K$ iterations, we have:

$$\frac{1}{K} \sum_{t=1}^{K} \mathbb{E}||\nabla F(\bar{\mathbf{x}}^{(t)})||^2 = O(\frac{n}{K} \frac{\rho}{(1 - \sqrt{\rho})^2} + \frac{1}{\sqrt{nK}}), \qquad (3)$$

where $\rho$ represents the largest singular value of matrix $W - \mathbf{1}\mathbf{1}^T/n$, $\mathbf{1} = [1, 1, ..., 1]_{n \times 1}^T$.

Based on Lemma 1, the convergence rate of the DecenSGD algorithm depends on three key factors: the number $n$ of nodes, iterations $K$, and the largest eigenvalue of $W - \mathbf{1}\mathbf{1}^T/n$. It is worth noting that a smaller $\rho$ leads to a reduced convergence error bound and a higher convergence rate. Thus, our investigation focuses on the manipulation of $W$ to achieve a decrease in $\rho$.

In order to guarantee the consensus among the nodes, we set $W$ as a symmetric and doubly stochastic mixing matrix (i.e., the sum of each column/row is 1) [4, 31]. In the model aggregation, a common choice is equal weight for each neighboring nodes, defined as follows:

$$W = I - \alpha L, \qquad (4)$$

where $\alpha$ is a small positive number and $L = D - A$ is the Laplacian matrix, $D = diag\{d_1, d_2, ..., d_n\}$ with $d_i$ being the degree of node $i$. $W$ satisfies symmetric and doubly-stochastic properties and is widely used in previous works [4, 31]. $\rho$ is simplified as

$$\rho = max\{(1 - \alpha\lambda_n))^2, (1 - \alpha\lambda_2)^2\}, \qquad (5)$$

where $\lambda_1 = 0 < \lambda_2 < ... < \lambda_n$ is the eigenvalues of matrix $L$ and the corresponding eigenvector is $\mathbf{v}_i$, $|\mathbf{v}_i| = 1$. Given a connected graph, there exists a trival eigenvalue $\lambda_1 = 0$ and the corresponding eigenvector $\mathbf{v}_1 = [1, 1, ..., 1]^T$. In practical scenarios, $\rho$ is usually small and $\rho$ is usually characterized by $(1 - \alpha\lambda_2)^2$ [4, 31]. Hence, a larger $\lambda_2$ will result in a smaller $\rho$ and better convergence rate. Our concern is how to choose a set of edges(denoted as $\mathcal{S}$) that could maximize $\lambda_2(L(\mathcal{S}))$, where $L(\mathcal{S})$ is the laplacian matrix after we add the set $\mathcal{S}$ of edges to the graph, abbreviated as $\lambda_2(\mathcal{S})$.

**Problem 1.** Computing the optimal set $\mathcal{S}$ of edges.

*Given:* The adjacency matrix of a connected graph and a budget number $k$.

*Find:* A budget number $k$ of edges $\mathcal{S}$ ($|\mathcal{S}| = k$ ) that, when added, could maximize the second smallest eigenvalue $\lambda_2(\mathcal{S})$ of the Laplacian matrix $L(\mathcal{S})$.

**Remark 1.** Maximizing $\lambda_2(\mathcal{S})$ could increase $\rho$, thereby improving the convergence rate. A parallel challenge arises when determining the optimal set of redundancy edges to be *removed* that don't influence $\lambda_2(\mathcal{S})$. The key issue of *Problem 1* is to characterize the importance of edges, which could also be used to identifying the insignificant edges (i.e., edges to be removed) with minor modification. Hence, in the method, we focus on the analysis of *adding* edges. In the experiment, we also present the results of *removing* edges.

# 4 THE PROPOSED ACCELERATING ALGORITHM

## 4.1 Complexity of $\lambda_2(\mathcal{S})$ optimization

we first consider five types of graphs in Fig. 1: star, line, circle, sparse graph, and fully connected graph. Fully connected graph has the same performance with the central-based FL, because every node has the same updating scheme with the parameter server in central-based FL. We observe that certain graphs may possess varying numbers of edges while sharing the same $\lambda_2$, implying that certain edges contribute insignificantly to the convergence rate of DFL.

| $\lambda_2 = 1$ | $\lambda_2 = 0.268$ | $\lambda_2 = 1$ | $\lambda_2 = 1$ | $\lambda_2 = 6$ |

**Figure 1: An example of $\lambda_2$ in different types of graphs. Notice that some graphs may have different number of edges but have the same $\lambda_2$.**

We consider the variation of $\lambda_2$ (eigen-var), $\Delta(\mathcal{S}) = \lambda_2(\mathcal{S}) - \lambda_2$, where $\lambda_2(\mathcal{S})$ denotes the second smallest eigenvalue of the Laplacian matrix after adding edges $\mathcal{S}$ to the graph. Problem 1 is rephrased of finding the optimal set $\mathcal{S}$ that maximizes $\Delta(\mathcal{S})$. We have the following theorem:

**Theorem 1.** Finding the best set $\mathcal{S}$ of added edges to maximize $\Delta(\mathcal{S})$ is NP-Complete.

*Proof*: We transfer the problem into the eigenvalue optimization of adjacency matrix. Let $Q = I - \delta L$, where $\delta$ is a small positive number, $\delta < \frac{1}{d_{max}}$ with $d_{max}$ being the largest degree of the graph. We have $Q_{ij} = \delta a_{ij}$, $i \neq j$, and otherwise $Q_{ij} = 1 - \delta d_i$, $i = j$. Hence, we could treat $Q$ an adjacency matrix of a graph that contain selfloops. $Q$ has a trival (largest) eigenvalue 1. $\lambda_2$ could be calculated as $\lambda_2 = (1 - \lambda_{n-1}(Q))/\delta$, where $\lambda_{n-1}(Q)$ is the second largest eigenvalue of $Q$. Hence, maximizing $\Delta(\mathcal{S})$ is equivalent to minimizing $\lambda_{n-1}(Q)$. Based on refs. [9, 38, 45], the eigenvalue minimization based on edge manipulation has been proved to be NP-Complete. Hence, we arrive at Theorem 1.□

For NP-Complete problem, it is impossible to calculate the optimal solution in polynomial time. Thus, we propose an approximation solution to solve the problem.

## 4.2 Approximation of $\lambda_2(\mathcal{S})$

After adding the set of edges $\mathcal{S}$ to the graph, we assess the eigenvalue of $\lambda_2(\mathcal{S})$ as

$$\widetilde{\lambda}_2(\mathcal{S}) = \lambda_2 + \sum_{(i,j)\in\mathcal{S}} (\mathbf{v}_{2,i} - \mathbf{v}_{2,j})^2, \qquad (6)$$

where $\mathbf{v}_{2,i}$ is the $i-$th entry of the eigenvector $\mathbf{v}_2$ and $\widetilde{\lambda}_2(\mathcal{S})$ means the estimation of $\lambda_2(\mathcal{S})$. Intuitively, based on Eq, 6, the addition of an edge has high 'eigen-var' if the endpoints of the edge are dissimilar with either. Adding edges to the

graph increases $\widetilde{\lambda}_2(\mathcal{S})$ and removing existing edges might decrease $\widetilde{\lambda}_2(\mathcal{S})$.

**Lemma 2**. Let $\sigma = \lambda_3 - \lambda_2$ denote the eigen-gap between the second and third smallest eigenvalues of $L$. If $\sigma >= 2\sqrt{2k^2 + 2k}$, then

$$\Delta(\mathcal{S}) = \sum_{(i,j)\in\mathcal{S}} (\mathbf{v}_{2,i} - \mathbf{v}_{2,j})^2 + O(2k^2 + 2k), \qquad (7)$$

where $\Delta(\mathcal{S}) = \lambda_2(\mathcal{S}) - \lambda_2$.

*Proof*: We first construct a perturbed matrix $E \in \mathbb{R}^{n\times n}$ after adding edges set $\mathcal{S}$. For the off-diagonal elements, $E_{ij} = -1$ if $(i,j) \in \mathcal{S}$; $E_{ij} = 0$ otherwise. For the diagonal elements, $E_{ii} = -\sum_{j=1:n,j\neq i} E_{ij}$. $E$ could be decomposed of $k$ sub-matrices, $E = E_1 + E_2 + ... + E_k$, where each $E_i$ represents the corresponding perturbed matrix of one edge in $\mathcal{S}$. Let the $i-$th edge in $\mathcal{S}$ has two endpoints $(a, b)$, we have

$$\mathbf{v}_2^T E_i \mathbf{v}_2 = \mathbf{v}_{2,a}^2 + \mathbf{v}_{2,b}^2 - 2\mathbf{v}_{2,a}\mathbf{v}_{2,b} = (\mathbf{v}_{2,a} - \mathbf{v}_{2,b})^2. \qquad (8)$$

According to the matrix perturbation theory [46], we have

$$\begin{aligned}\lambda_2(\mathcal{S}) &= \lambda_2 + \mathbf{v}_2^T E \mathbf{v}_2 + O(||E||_F^2) \\ &= \lambda_2 + \sum_{i=1:k} \mathbf{v}_2^T E_i \mathbf{v}_2 + O(||E||_F^2) \\ &= \lambda_2 + \sum_{(a,b)\in\mathcal{S}} (\mathbf{v}_{2,a} - \mathbf{v}_{2,b})^2 + O(||E||_F^2), \qquad (9)\end{aligned}$$

where $||E||_F$ is the *Frobenious norm* of $E$, $||E||_F = \sqrt{\sum_i \sum_j E_{ij}^2}$. Recalling the property of $E$ that there are $2k$ off-diagonal elements with value $-1$, and the sum of the diagonal elements of $E$ is 2k, hence we have $4k \leq ||E||_F^2, ||E||_2^2 \leq 2k^2 + 2k$, where $||E||_2$ is the $l_2$ *norm* of $E$. Moreover, based on the matrix perturbation theory [46], we also have

$$\widetilde{\lambda}_2(\mathcal{S}) \leq \lambda_2 + ||E||_2 \leq \lambda_2 + \sqrt{2k^2 + 2k}, \qquad (10)$$

$$\widetilde{\lambda}_i(\mathcal{S}) \geq \lambda_i - ||E||_2 \geq \lambda_i - \sqrt{2k^2 + 2k}, i = 3, 4, ..., n. \qquad (11)$$

Since $\sigma >= 2\sqrt{2k^2 + 2k}$, we have $\widetilde{\lambda}_2(\mathcal{S}) < \widetilde{\lambda}_i(\mathcal{S}), i = 3, 4, ..., n$. Hence $\widetilde{\lambda}_2(\mathcal{S})$ is the estimation of $\lambda_2(\mathcal{S})$ and we arrive at

$$\Delta(\mathcal{S}) = \sum_{(i,j)\in\mathcal{S}} (\mathbf{v}_{2,i} - \mathbf{v}_{2,j})^2 + O(2k^2 + 2k), \qquad (12)$$

which completes the proof.□

Lemma 2 provides a convenient and efficient approach for the rapid evaluation of the eigen-var, instead of recalculating the corresponding eigenvalue. In the subsequent section, we leverage Lemma 2 to devise a fast algorithm aimed at maximizing $\lambda_2(\mathcal{S})$.

## 4.3 Proposed algorithm SSEO

The propose **S**econd **S**mallest **E**igenvalue **O**ptimization (SSEO) algorithm is shown in Alg. 1. The central issue of Alg. 1 is to choose the best set of edges to maximize $\widetilde{\lambda}_2(\mathcal{S})$ in Eq. 6.

In Algorithm 1, the Laplacian matrix $L$ (line 1) and the second smallest eigen-pair $(\lambda_2, \mathbf{v}_2)$ (line 2) are computed. We initialize the set $\mathcal{S}$ as empty (line 3) and use a temporary array $U$ to store the values of each potential edge (line 4). The values of each nonexisting edge are calculated in lines

5-14. The top-$k$ edges in $U$ are determined in lines 14-15 using a sorting algorithm and the selected edges are then stored in $\mathcal{S}$.

---

**Algorithm 1:** Algorithm to optimize $\lambda_2(\mathcal{S})$ (SSEO)

---

**Input**: The adjacency matrix $A$, and a budget number $k$.
**Output**: Edge set $\mathcal{S}$ with $k$ elements.
**1** Compute the Laplacian matrix $L = D - A$;
**2** Compute the second smallest eigenvalue $\lambda_2$ and the corresponding eigenvector $\mathbf{v}_2$;
**3** Initialize $\mathcal{S}$ to be empty;
**4** initialize an array $U[n][n]$; //Each element corresponds to one potential edge.
**5** **for** $i = 1 : n$ **do**
**6**  **for** $j = i + 1 : n$ **do**
**7**   **if** $A[i][j] == 1$ **then**
**8**    $U[i][j] = 0$;
**9**   **else**
**10**    $U[i][j] = (\mathbf{v}_{2,i} - \mathbf{v}_{2,j})^2$;
**11**   **end**
**12**  **end**
**13** **end**
**14** Get the top-$k$ elements from the uppertriangular part of $U$ by the descending order and save the values and corresponding edges to array $R$;
**15** Add edges in $R$ to $\mathcal{S}$.
**16** return $\mathcal{S}$;

---

Next, we analyze the accuracy and efficiency of SSEO.

**Theorem 2 Effectiveness of SSEO:** Let $\widetilde{\Delta}(\mathcal{S}) = \sum_{(i,j)\in\mathcal{S}} (\mathbf{v}_{2,i} - \mathbf{v}_{2,j})^2$. Alg. 1 could maximize the $\widetilde{\Delta}(\mathcal{S})$.

*Proof*: Let $\mathcal{S}$ and $\mathcal{S}^*$ denote the sets detemined by Alg. 1 and the theoretical best set, respectively, and the corresponding eigenvalue variances are $\widetilde{\Delta}(\mathcal{S})$ and $\widetilde{\Delta}(\mathcal{S}^*)$.

We consider two cases:

Case 1: $\mathcal{S} \bigcap \mathcal{S}^* = \emptyset$. In lines 14–15, we chosen edges with top-$k$ values, and hence $\forall (a, b) \in S$ and $\forall (c, d) \in S^*$, $(\mathbf{v}_{2,a} - \mathbf{v}_{2,b})^2 \geq (\mathbf{v}_{2,c} - \mathbf{v}_{2,d})^2$. Consequently, $\widetilde{\Delta}(\mathcal{S}) \geq \widetilde{\Delta}(\mathcal{S}^*)$. Since $\mathcal{S}^*$ is the best set, we also have $\widetilde{\Delta}(\mathcal{S}) \leq \widetilde{\Delta}(\mathcal{S}^*)$. As a result, $\widetilde{\Delta}(\mathcal{S}) = \widetilde{\Delta}(\mathcal{S}^*)$.

Case 2: $H = \mathcal{S} \bigcap \mathcal{S}^* \neq \emptyset$. $\widetilde{\Delta}(\mathcal{S})$ and $\widetilde{\Delta}(\mathcal{S}^*)$ could be decomposed as $\widetilde{\Delta}(\mathcal{S}) = \widetilde{\Delta}(\mathcal{H}) + \widetilde{\Delta}(\mathcal{S} \setminus \mathcal{H})$. $\widetilde{\Delta}(\mathcal{S}^*) = \widetilde{\Delta}(\mathcal{H}) + \widetilde{\Delta}(\mathcal{S}^* \setminus \mathcal{H})$. Based on case 1, we also have $\widetilde{\Delta}(\mathcal{S} \setminus \mathcal{H}) = \widetilde{\Delta}(\mathcal{S}^* \setminus \mathcal{H})$. Hence, $\widetilde{\Delta}(\mathcal{S}) = \widetilde{\Delta}(\mathcal{S}^*)$.

Combing the two cases, we have $\widetilde{\Delta}(\mathcal{S}) = \widetilde{\Delta}(\mathcal{S}^*)$. $\square$

**Lemma 3 Time complexity of SSEO.** The computational complexity of Alg. 1 is $O(n^2 + k\ln(n))$.

*Proof*: Line 1 costs time $O(n)$. Computing the eigen-pair $(\lambda_2, \mathbf{v}_2)$ (line 2) costs $O(|E|)$. Lines 3–4 cost $O(1)$. Lines 5–13 cost $O(n^2)$. Lines 14–15 could be executed by heap sort that has time complexity $O(k\ln(n^2))$. Hence, the overall time complexity is $O(n^2 + k\ln(n))$.$\square$

**Lemma 4 Space cost of SSEO.** The space cost of Alg. 1 is $O(n^2)$.

*Proof*: The space of $L$ is $O(n^2)$ in line 1. In line 4, $U$ costs space $O(n^2)$. In line 14, we could use a temporary array to save the sorted top-$k$ values, which costs $O(n^2)$. Hence the overall space cost is $O(n^2)$.$\square$

---

**Algorithm 2:** Algorithm to optimize $\lambda_2(\mathcal{S})$ (SSEO+)

---

**Input**: The adjacency matrix $A$, an integer $r$, and a budget number $k$.
**Output**: Edge set $\mathcal{S}$ with $k$ elements.
**1** $\mathcal{S} = SSEO(A, k)$;
**2** Compute the Laplacian matrix $L = D - A$;
**3** Compute the smallest-$r$ eigenvalue $\lambda_i$ and the corresponding eigenvector $\mathbf{v}_i$ $(i = 1, 2, ..., r)$;
**4** Calculate $\widetilde{\lambda}_i(\mathcal{S})$ based on Eq. 13;
**5** initialize an array $U[n][n]$;
**6** **for** $i=1:k$ **do**
**7**  Let $(a, b)$ be the $i-$th edge in $\mathcal{S}$ and compute $\zeta(\{(a, b)\})$ based on Eq. 15;
**8**  Remove $(a, b)$ from $\mathcal{S}$;
**9**  **for** $i = 1 : n$ **do**
**10**   **for** $j = i + 1 : n$ **do**
**11**    **if** $A[i][j] == 1$ **then**
**12**     $U[i][j] = 0$;
**13**    **else**
**14**     $\zeta(\{(i,j)\}) = \widetilde{\lambda}_2(\mathcal{S} \bigcup \{(i,j)\}) - \widetilde{\lambda}_2(\mathcal{S})$;
**15**     $U[i][j] = \zeta(\{(i,j)\})$;
**16**    **end**
**17**   **end**
**18**  **end**
**19**  Compute the largest elements from $U$, denoted as $\zeta_{max}$ and the corresponding edge is $(c, d)$;
**20**  **if** $\zeta(\{(a, b)\}) \geq \zeta_{max}$ **then**
**21**   Add $(a, b)$ to $\mathcal{S}$;
**22**  **else**
**23**   Add $(c, d)$ to $\mathcal{S}$;
**24**  **end**
**25** **end**
**26** return $\mathcal{S}$;

---

## 4.4 A Variant: SSEO+ Algorithm

In Lemma 2, it is required that the eigen-gap and the budget number $k$ satisfy $\sigma \geq 2\sqrt{2k^2 + 2k}$. Consequently, we have $k \leq k_{max} = \sqrt{\frac{\sigma}{8} + \frac{1}{4}} - \frac{1}{2}$. This constraint implies that in order to obtain a high approximation of $\lambda_2(\mathcal{S})$, the number of candidate edges should be less than $k_{max}$. However, this constraint cannot be met when we aim to add more edges. It is worth noting that $\sigma$ is typically small in real graphs. Previous research in the field of graph community detection has shown that the Laplacian matrix of a graph has $r - 1$ eigenvalues close to zero, where $r$ is the number of communities. Real-world graphs such as Internet AS graphs, online

social graphs, and infrastructure graphs often exhibit community structure. Consequently, $\sigma$ is usually small, rendering Algorithm 1 inappropriate for large $k$.

To address the problem, we propose SSEO+ algorithm in Alg. 2. In Alg. 2, we estimate the $\widetilde{\lambda}_2(\mathcal{S})$ as

$$\widetilde{\lambda}_2(\mathcal{S}) = min\{\lambda_i + \sum_{(a,b) \in \mathcal{S}} (\mathbf{v}_{i,a} - \mathbf{v}_{i,b})^2, i = 2, 3, ..., r\}, \quad (13)$$

where $r$ is the number of communities in the graph. In fact, Eq. 6 is a particular case of Eq. 13 when $r = 2$. The objective is to choose the best $\mathcal{S}$ that maximizes $\widetilde{\lambda}_2(\mathcal{S})$ in Eq. 13, which is a combinational optimization problem, defined as

$$max_{\mathcal{S}}\{min\{\lambda_i + \sum_{(a,b) \in \mathcal{S}} (\mathbf{v}_{i,a} - \mathbf{v}_{i,b})^2, i = 2, 3, ..., r\}\}, \quad (14)$$

$$s.t.|\mathcal{S}| = k.$$

Optimizing the aboved combinational optimization problem is NP-hard. In Alg. 2, SSEO+ optimizes Eq. 14 by improving SSEO. After computing a candidate set $\mathcal{S}$ based on SSEO (line 1) in Alg. 2, we compute the contribution of each edge in $\mathcal{S}$ (line 7), where the contribution of edge $(i, j) \in \mathcal{S}$ is evaluated as

$$\zeta(\{(i,j)\}) = \widetilde{\lambda}_2(\mathcal{S}) - \widetilde{\lambda}_2(\mathcal{S} \setminus \{(i,j)\}), \quad (15)$$

where $\widetilde{\lambda}_2(\mathcal{S})$ is computed by Eq. 13. If the contribution of an edge $(i, j)$ in $\mathcal{S}$ is less than the ones $(i', j')$ in the remaining edge set, we remove $(i, j)$ from $\mathcal{S}$ and add $(i', j')$ to $\mathcal{S}$ (lines 6–25), where the contribution of $(i', j')$ is computed in lines 9–18.

By a similar procedure for SSEO, we can show that the time complexity of SSEO+ is $O(kn^2r)$; and its space cost is the same as that of SSEO.

**Determination of $r$**: There is a free parameter $r$ in Alg. 2. It is important to note that increasing the value of $r$ has the potential to enhance the performance, with the cost of more time consumption. Specifically, $r$ denotes the number of communities in the graph. We could first perform community detection to determine $r$. Numerous community detection techniques can be employed for this purpose. In the experiment, the *Louvain* method is utilized to determine the value of $r$, where *Louvain* is a well known modularity-based method to detect communities in graphs [47, 48].

**Remark 2.** SSEO and SSEO+ exclusively address the scenario of adding new edges, failing to consider the removal of redundant edges. However, it is possible to extend both methods to account for removing redundant edges. Specifically, in Alg. 1, we adapt line 14 to select the least-$k$ elements. In Alg. 2, we modify line 19 to compute the smallest element from set $U$. These modifications are relatively minor in nature. It is noteworthy that we also include the experimental results pertaining to the removal of edges based on the minor modifications.

## 5 EXPERIMENT

Here, we are interested in the performance of theoretical analysis on real data. Our experiments run on a cluster of four computers with 1 2.4GHz Intel(R) i7 CPU, 32GB memory and 64bit Ubuntu 20.04.

### 5.1 Experimental setup

**Datasets.** We conduct experiments on four real datasets. The four real datasets [1] include: (a) **RealityMining**: This is an undirected graph containing human contact data among students of the Massachusetts Institute of Technology (MIT), collected by the Reality Mining experiment performed in 2004 as part of the Reality Commons project. The graph has 96 nodes and 2539 edges. (b) **NetScience**: This is a graphs of co-authorships in the area of network science. The graph has 379 nodes and 914 edges. (c) **AS**: This is an undirected graph of autonomous systems of the Internet. Nodes are autonomous systems (AS), and edges denote communication. The graph has 487 nodes and 1078 edges. (d) **Facebookego**: The is the friendship of an ego graph on Facebook. The graph has 2888 nodes and 2981 edges.

**Environment of DFL.** On the computer cluster, we simulate the decentralized federated learning, where each node is represented by a deep learning model and the connections between nodes follow the structure introduced in the datasets. We evaluate the performance in the image classification on CIFAR-10 [49]. CIFAR-10 consists of 60000 color images in 10 classes. We use the classical ResNet-50 as the deep learning model to be trained. All images are evenly partitioned over all nodes. The initial learning rate is set as 0.8 and it decays by 10 after 100 epochs (The learning rate is fine-tuned for vanilla DecenSGD and used for all other algorithms). The mini-batch size is 64 and the model is trained at most 1500 epochs.

**Benchmark Methods.** We compare our methods SSEO and SSEO+ with four state-of-the-art methods.

- RW$^{Chaos2008}$ [39]: This method increases the synchronization in complex network by rewiring edges based on the graph Laplacian eigenvectors. .
- RatioW$^{TNSE2016}$ [6]: This method optimizes the synchronization in complex graphs based on the perturbation of the ratio of the second smallest eigenvalue and the largest eigenvalue of the Laplacian matrix.
- MATCHA$^{TSP2022}$ [5]: This method allows nodes to communicate more frequently over connectivity-critical edges to increase the synchronization speed to solve the bottleneck of communication delay.
- CoCo$^{TMC2023}$ [50]: This method accelerates DFL by optimize the weight of the existing edges and model compression. In the method, we allow the method to optimize a budget $k$ of edges as well as our methods.

**Evaluation metrics.** The central metric to evaluate the effectiveness of different methods is the second smallest eigenvalue $\lambda_2(\mathcal{S})$. Larger $\lambda_2(\mathcal{S})$ means more effectiveness of the methods and is better. Besides, we also compare the convergence rate of different methods and higher convergence rate is better. At last, we compare the time consumption under different graphs to evaluate the efficiency.

---

[1]Konect Network Collection. http://konect.cc/networks/

## 5.2 Experiment results

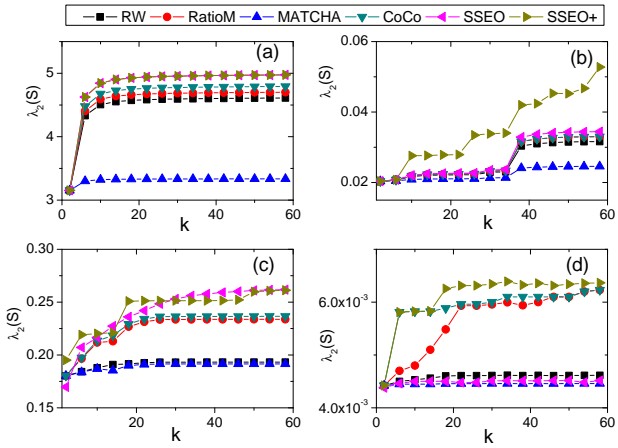

Figure 2: $\lambda_2(\mathcal{S})$ as a function of the number of added edges. Larger $\lambda_2(\mathcal{S})$ is better. (a) RealityMining graph. (b) NetScience graph. (c) AS graph. (d) Facebookego graph.

*5.2.1 Effectiveness of SSEO and SSEO+.* We compare the $\lambda_2(\mathcal{S})$ with respect to the number of added edges in Fig. 2. In Fig. 2, our proposed method SSEO+ outperforms the existing methods across all graph instances. We notice that MATCHA exhibits poor performance in Fig. 2, because MATCHA aims to optimize the communication frequencies on different edges to mitigate communication delays. The underlying graph structure remains unchanged by MATCHA, and hence, the variation in $\lambda_2(\mathcal{S})$ is relatively minimal. On the other hand, SSEO fluctuates largely in different graphs, SSEO performs well in Fig. 2(a)(c), but not so good in (b)(d). Because we assume that the eigen-gap between the second and third largest eigenvalues of $L$ is large in Lemma 2. However, some real graphs have small eigen-gaps that don't satisfy the assumption. Hence, SSEO exhibits poorer performance in such graph instances. Conversely, SSEO+ considers the case of small eigen-gaps, which greatly increases $\lambda_2(\mathcal{S})$.

Besides, we show the performance of SSEO and SSEO+ on removing redundancy edges in Fig. 3. We modify SSEO and SSEO+ by consider the least decrease of $\lambda_2(\mathcal{S})$ as discussed in *remark 2*. Figure 3 shows that removing the redundancy edges rarely influences the $\lambda_2(\mathcal{S})$ and the convergence rate, validating the effectiveness of our methods.

We then perform DFL using vanilla decentralized SGD on different graphs. Here, we perform vanilla decentralized SGD on two graphs (RealityMining and NetScience) and use the classical cross entropy as the loss function. Figure 4 shows the training loss evolution over epochs for the aforementioned graphs. We see that the training loss of SSEO+ decays (decreases) faster than the other methods, which agrees well with Fig. 2. To better capture the convergence rate, Table 1 displays the minimum number of epochs required for the training loss to drop below 0.1 for Fig. 4. We see that SSEO+

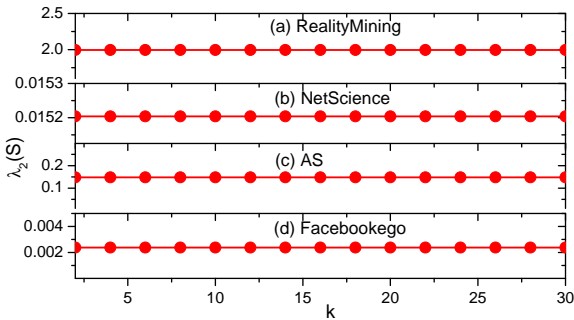

Figure 3: $\lambda_2(\mathcal{S})$ as a function of the number of removed (redundancy) edges. Stable $\lambda_2(\mathcal{S})$ is better.

Table 1: The minimum epoches that the training loss is less than 0.1. We train a ResNet-50 model on CIFAR-10 dataset. We count the epoches that the training loss first decreases to 0.1. Smaller is better.

| graph | RW | RatioW | CoCo | SSEO | SSEO+ |
|---|---|---|---|---|---|
| RealityMining | 255 | 254 | 255 | 232 | **232** ↓ |
| NetScience | 1490 | 1489 | 1055 | 1060 | **957** ↓ |

exhibits the lowest number of epochs, indicating a higher convergence rate. In Table 1, we don't show the minimum epochs for MATCH method, because MATCH requires much larger epochs to reach 0.1 in Fig. 4, which is omitted for space limitation. Besides, the performance of vanilla decentralized SGD in the other two graphs are similar to Fig. 4 and Table 1, which isn't show to save space.

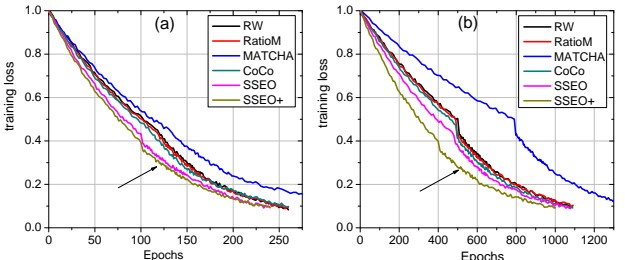

Figure 4: The training loss as a function of epochs in different graphs. We train a ResNet-50 model on CIFAR-10 dataset. Each subfigure represents a result using different graphs as the structure of DFL. (a) RealityMining graph. (b) NetScience graph.

*5.2.2 Efficiency of SSEO and SSEO+.* Since SSEO is actually a specific case that focuses only on perturbing the second smallest eigenvalue, we primarily investigates the time complexity of SSEO+ in graphs with varying numbers of nodes and edges. We first use the configuration model [7] to generate random graphs with different numbers of nodes. Figure 5 illustrates the time consumption as a function of the number

**Table 2: The pearson correlation between estimated eigenvalue $\widetilde{\lambda}_2(\mathcal{S})$ and real eigenvalue $\lambda_2(\mathcal{S})$ for different budget $k$ of added edges.**

| k | 'RealityMining' | 'NetScience' | 'AS' | 'Facebookego' |
|---|---|---|---|---|
| 2 | 0.9995 | 0.9993 | 0.9997 | 0.9991 |
| 5 | 0.9991 | 0.9952 | 0.9991 | 0.9982 |
| 10 | 0.9972 | 0.9902 | 0.9634 | 0.9908 |
| 20 | 0.9945 | 0.9150 | 0.9837 | 0.9639 |
| 40 | 0.9989 | 0.7591 | 0.9805 | 0.8694 |

of nodes and edges. Figure 5(a) demonstrates that graphs with the same number of edges and varying numbers of nodes exhibit a quadratic increase in time consumption. Conversely, Figure 5(b) shows that graphs with the same number of nodes but different numbers of edges have negligible changes in time consumption, as the time complexity is solely dependent on the number of nodes.

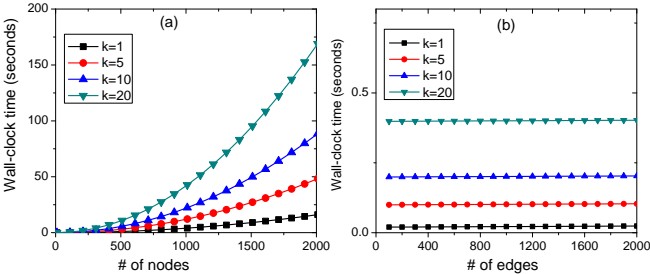

**Figure 5: (a) Time consumption as a function of the number of nodes in the graph. All graphs have 5000 edges, but different numbers of nodes. (b) Time consumption as a function of the number of edges in the graph. All graphs have 100 nodes, but different numbers of edges.**

*5.2.3 Validation of our assumption.* In Eq. 6, we evaluate the eigenvalue $\widetilde{\lambda}_2(\mathcal{S})$ based on the first-order perturbation of $\lambda_2$. Table 2 presents the Pearson correlation between $\widetilde{\lambda}_2(\mathcal{S})$ and $\lambda_2(\mathcal{S})$ for different budget $k$ of added edges. When $k$ is small, $\widetilde{\lambda}_2(\mathcal{S})$ approximates $\lambda_2(\mathcal{S})$ quite well; with the increase of $k$, the Pearson correlation is still high ($\geq 0.8$). Note that when $k = 40$, the Pearson correlation is less than 0.8 in the NetScience graph, because the $\lambda_2(\mathcal{S})$ is determined by the perturbation of other eigenvalues under the scenario, which is also reflected in Fig. 2. If we use Eq. 13 to evaluate $\lambda_2(\mathcal{S})$, the Pearson correlation is still larger than 0.8.

*5.2.4 A case study in real graph.* Our methods SSEO and SSEO+ offer a solution for selecting influential edges to enhance $\lambda_2(\mathcal{S})$, as well as identifying redundant edges that can be removed without decreasing $\lambda_2(\mathcal{S})$.

We present a graphical representation (see Fig. 6) depicting the addition of edges (highlighted in red) and the removal of edges (highlighted in green) in the RealityMining graph

based on the SSEO+. In Fig. 6, the added edges tend to connect nodes with distinct entry values in the eigenvector, whereas the removed edges are inclined to connect nodes with similar entry values. These findings are consistent with the analysis of Eq. 6 that the contribution of each edge is $(\mathbf{v}_{2,i} - \mathbf{v}_{2,i})^2$.

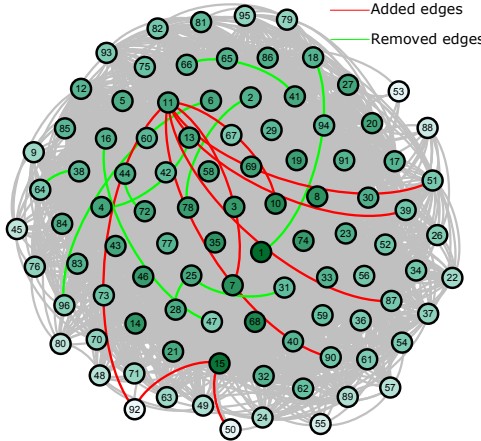

**Figure 6: A schematic illustration of added edges (red) and removed edges (green) in RealityMining graph. The color depth of nodes represents the entry value in the eigenvector $\mathbf{v}_2$. The added edges are inclined to connect nodes with quite different color depth; while the removed edges are inclined to connect nodes with similar color depth.**

## 6 CONCLUSION

In this paper, we address the issue of accelerating the Distributed Federated Learning by augmenting the communication graph with additional edges. Firstly, we formally define the convergence rate of DFL and demonstrate that increasing the second smallest eigenvalue of the Laplacian matrix associated with the communication graph can enhance the convergence rate. Subsequently, we investigate the problem of how to choose edges that maximize the second smallest eigenvalue. Through our analysis, we quantify the perturbation in eigenvalues caused by the addition of new edges. Furthermore, we develop effective algorithms to select the optimal edges for maximizing the second smallest eigenvalue. The experimental results clearly indicate that our algorithm surpasses existing methods by a substantial margin.

Our method could be easily combined with existing accelerating methods, such as parameter compression and communication scheduling. Besides, it is worth mentioning that our approach can be extended to address the removal of such insignificant edges. Within the communication framework of DFL, certain edges may have negligible impact on the convergence rate, but consume communication bandwidth. Further investigation is still required to uncover the underlying mechanisms governing redundant edges.

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

Received 20 February 2007; revised 12 March 2009; accepted 5 June 2009

