# OpenReview forum: "Accelerating the decentralized federated learning via manipulating edges"
_ACM.org/TheWebConf/2024/Conference — TheWebConf24_

### Official Review · Reviewer_Mqst · 2023-11-22

**Novelty:** 4
**Technical Quality:** 6

**Review:**

Decentralized federated learning (DFL) enhances reliability and security through peer-to-peer (P2P) model sharing. However, DFL encounters challenges with slow convergence due to complex P2P graphs. To expedite DFL, they introduce an efficient algorithm by adding a limited number 𝑘 of edges to P2P graphs. They establish a connection between convergence rate and the second smallest eigenvalue of the Laplacian matrix, proving that finding the optimal edge set is NP-complete. Their analysis demonstrates the positive impact of strategic edge additions on this eigenvalue. They propose an efficient 𝑂(𝑘𝑟𝑛2) algorithm to compute the best edge set, maximizing the second smallest eigenvalue and thus convergence rate. Experimental results across diverse datasets validate the effectiveness of their approach in accelerating DFL convergence.

Strengths:
1.	The theoretical analysis is solid.

2.	The paper is easy to follow.


Weakness:
1.	The novelty is limited. As far as I know, there has been many similar DML works that consider the topology, e.g., [1,2].

[1] Giovanni Neglia, Gianmarco Calbi, Don Towsley, Gayane Vardoyan: The Role of Network Topology for Distributed Machine Learning. INFOCOM 2019: 2350-2358

[2] Angelia Nedic, Alex Olshevsky, Michael G. Rabbat:
Network Topology and Communication-Computation Tradeoffs in Decentralized Optimization. Proc. IEEE 106(5): 953-976 (2018)

2.	The number of edges k is hard to determine in practical settings.

3.	It is hard for multiple decentralized clients to collaboratively and dynamically decide the topology in practical settings, especially when the training time in each round is usually small of DML.

**Questions:**

What are the hardware configurations for the simulation evaluations?

**Reviewer Confidence:**

3: The reviewer is confident but not certain that the evaluation is correct

**Scope:**

4: The work is relevant to the Web and to the track, and is of broad interest to the community

---

### Official Review · Reviewer_4diC · 2023-11-23

**Novelty:** 4
**Technical Quality:** 5

**Review:**

Summary: This paper proposed an algorithm to accelerate Decentralized Federated Learning (DFL) by introducing a limited number k of edges into the peer-to-peer graphs.

Strength:
1. This paper shows that accelerating DFL could be achieved by increasing the second smallest eigenvalue of the Laplacian matrix associated with a graph.
2. The proposed algorithm remarkably reduced the operation time complexity.
3. Three benchmarks and four baselines demonstrate the effectiveness of the proposed algorithm.

**Questions:**

Can the proposed FL framework address the data heterogeneous problem? Can the model still converge under this case?

**Ethics Review Description:**

N.A.

**Reviewer Confidence:**

2: The reviewer is willing to defend the evaluation, but it is likely that the reviewer did not understand parts of the paper

**Scope:**

3: The work is somewhat relevant to the Web and to the track, and is of narrow interest to a sub-community

---

### Official Review · Reviewer_1Fp4 · 2023-11-23

**Novelty:** 5
**Technical Quality:** 5

**Review:**

This work proposes two approaches (SSEO and SSEO+) for accelerating decentralized federated learning by manipulating the edges of the existing P2P graphs. They evaluated their approach using open state-of-the-art network datasets when simulating the P2P connections of the network.

The theorems look reasonable, but need to say that they are beyond my expertise, so I will trust the other reviewers on this part.

Some strong points of this work:
+ Novel and original work
+ Appreciated the complexity calculation
+ Very well-written and detailed manuscript.
+ Detailed related work, covering the areas of FL, DFL, Edge manipulation etc.

Some points that could be improved:
- While I do appreciate the network datasets you have used to evaluate your algorithms, all of your evaluation depends on a single image recognition dataset (CIFAR-10). I agree this is a standard and very popular dataset in ML, however, at the same time, it is limited to the scope of computer vision.
- One of the main challenges of FL (and consequently, DFL) is the Non-IID distribution of the samples. In your work, you limit your evaluation in an IID setting (i.e., "All images are evenly partitioned over 664 all nodes"). I would have expected to experiment with, either a pre-set NonIID split of the samples (or even better use different alpha from Dirichlet distributions), or use a dataset with an existing non-iid distribution (e.g., Shakespeare).
- You report the specifications of the machines you used to run your experiments. Considering you are not conducting any performance evaluations, I can't see why this is relevant. Better to report the frameworks you have used, versions etc so the results can be reproducible.
- Some background is missing, e.g., describing what eigenvalue is in networks.

**Questions:**

- How do you think the results would be affected when applying your approach to different types of models/datasets?
- Would different degrees of Non-IID have an impact on the performance?

**Ethics Review Description:**

No ethical issues considering the datasets used are public

**Reviewer Confidence:**

3: The reviewer is confident but not certain that the evaluation is correct

**Scope:**

4: The work is relevant to the Web and to the track, and is of broad interest to the community

---

### Official Review · Reviewer_aLb2 · 2023-11-25

**Novelty:** 5
**Technical Quality:** 5

**Review:**

Federated learning (FL) enables multiple clients to jointly train a global machine learning model. Existing FL approaches depend on a central server, which can lead to communication and computational bottlenecks as the number of agents increases. To tackle these issues, Distributed Federated Learning (DFL) has been introduced. Nevertheless, DFL encounters challenges related to its slow convergence rate. In this paper, the authors present an efficient algorithm aimed at accelerating DFL.

Strengths:
1) The paper is well written and presents its ideas in a clear and comprehensible manner.
2) Thorough experiments provide strong evidence for the effectiveness of the proposed approach.

Weaknesses:
1) This paper appears to focus on optimization, which may not align well with The Web Conference. The authors have not adequately addressed the relevance of their work to the conference's scope.
2) The paper exclusively utilizes the CIFAR-10 dataset. It would be beneficial for the authors to replicate their experiments using a broader range of datasets from various domains, particularly those related to web applications.

**Questions:**

Please answer the weakness in the review.

**Reviewer Confidence:**

2: The reviewer is willing to defend the evaluation, but it is likely that the reviewer did not understand parts of the paper

**Scope:**

2: The connection to the Web is incidental, e.g., use of Web data or API

---

### Official Review · Reviewer_TEtw · 2023-11-26

**Novelty:** 5
**Technical Quality:** 5

**Review:**

This paper proposed a network topology optimization method that maximize the second largest eigen value of the graph Laplacian to improve convergence rate of the decentralized federated learning. The authors conducted several experiments on several real-world graphs, validating the effectiveness of the proposed algorithms.

However, this paper is not well-motivated. Adding edges in real-world scenarios is not usually practical. It is not reasonable to add an edge in a citation network if there is no citation relationship between the two authors, and it is not reasonable to add a link in an ad-hoc network if the two devices are not nearby to each other. The authors did not provide persuasive motivation examples in this paper to prove the rationality of the proposed methods.

Some of the notations may be misleading. In line 387, $\lambda_{n-1}(Q)$ is used to represent the second largest eigenvalue of Q. However, similar notations is used in line 331 with different meanings.

We suggest the authors to provide some theoretical analysis on the approximation rate of SSEO and SSEO+, since they are heuristic algorithms to solve the NP-complete problem approximately.

The evaluation of this paper is not enough. Although the authors evaluate the proposed method on several graphs, only the CIFAR-10 training data set is used. Besides, it is not reasonable to train a CIFAR-10 classification model on a social network or a citation network.

In the case study, we suggest the authors provide more straightforward illustration instead of the color depth, for example, the diameter of the graph.

**Questions:**

Are there any real-world examples proving that add edges in a graph is feasible?

Since there is not a central server in the decentralized learning network to coordinate the training process, what device is in the charge of edge manipulation? Can it know the graph topology exactly when the graph is large?

In both figure of the Fig. 4, why there is a drop of training loss around 0.4~0.6?

In Fig. 2 (b), (c) and (d), why there are some sudden improvements of $\lambda_2(S)$ when $k$ increases?

Where does Eq. 6 come from? How well is this approximation?

**Reviewer Confidence:**

2: The reviewer is willing to defend the evaluation, but it is likely that the reviewer did not understand parts of the paper

**Scope:**

3: The work is somewhat relevant to the Web and to the track, and is of narrow interest to a sub-community

---

### Decision · Program_Chairs · 2024-01-22

**Decision:**

Accept

**Comment:**

This paper proposed a network topology optimization method, which maximizes the second-largest eigenvalue of the graph Laplacian to improve the convergence rate of decentralized federated learning. Specifically, a limited number of edges is introduced to the peer-to-peer graphs.

 # Strengths:

 1 **Theoretical analysis** This paper provides detailed theoretical analysis proving that finding the optimal set of edges to maximize this eigenvalue is an NP-complete problem.

 2 **Well-written** Most reviewers agree that this paper is well-written with sufficient details.

 # Weaknesses:

 1 **Vague motivation** Persuasive motivation example of adding edge is missing in the original manuscript.

 2 *Novelty* Several important references are missing. The original manuscript lacks a comparison between them.

 3 **Insufficient experiments** As pointed out by several reviewers, the original manuscript did not consider a non-iid setting in FL (added during the rebuttal period) and was only conducted on the CIFAR-10 dataset.